# Mitochondrial Dysfunction and Therapeutic Perspectives in Cardiovascular Diseases

**DOI:** 10.3390/ijms232416053

**Published:** 2022-12-16

**Authors:** Yu Liu, Yuejia Huang, Chong Xu, Peng An, Yongting Luo, Lei Jiao, Junjie Luo, Yongzhi Li

**Affiliations:** 1China Astronaut Research and Training Center, Beijing 100094, China; 2Department of Nutrition and Health, China Agricultural University, Beijing 100193, China

**Keywords:** cardiovascular disease, mitochondrial dysfunction, mitochondrial DNA mutation, mitophagy, oxidative phosphorylation, reactive oxygen species, therapeutic strategy

## Abstract

High mortality rates due to cardiovascular diseases (CVDs) have attracted worldwide attention. It has been reported that mitochondrial dysfunction is one of the most important mechanisms affecting the pathogenesis of CVDs. Mitochondrial DNA (mtDNA) mutations may result in impaired oxidative phosphorylation (OXPHOS), abnormal respiratory chains, and ATP production. In dysfunctional mitochondria, the electron transport chain (ETC) is uncoupled and the energy supply is reduced, while reactive oxygen species (ROS) production is increased. Here, we discussed and analyzed the relationship between mtDNA mutations, impaired mitophagy, decreased OXPHOS, elevated ROS, and CVDs from the perspective of mitochondrial dysfunction. Furthermore, we explored current potential therapeutic strategies for CVDs by eliminating mtDNA mutations (e.g., mtDNA editing and mitochondrial replacement), enhancing mitophagy, improving OXPHOS capacity (e.g., supplement with NAD^+^, nicotinamide riboside (NR), nicotinamide mononucleotide (NMN), and nano-drug delivery), and reducing ROS (e.g., supplement with Coenzyme Q10 and other antioxidants), and dissected their respective advantages and limitations. In fact, some therapeutic strategies are still a long way from achieving safe and effective clinical treatment. Although establishing effective and safe therapeutic strategies for CVDs remains challenging, starting from a mitochondrial perspective holds bright prospects.

## 1. Introduction

The cardiovascular system plays a crucial role in the normal metabolism of the human body, also known as the circulatory system, which consists of arteries, veins, the heart, and capillaries. Common cardiovascular diseases (CVDs) include heart related heart failure, myocardial hypertrophy, arterial related atherosclerosis, aortic dissection, abdominal aortic aneurysm and other diseases [1]. CVDs constitute a leading worldwide health problem and account for a high proportion of global deaths, with a mortality rate of up to 20% [2]. Thus, it is imperative to explore the pathogenesis of CVDs and develop effective therapeutic strategies.

In fact, mitochondrial dysfunction is considered to be one of the important mechanisms affecting the pathogenesis of CVDs [3,4,5]. Mitochondria are key double-membrane organelles for aerobic respiration in biological cell, and generate energy required by cells through OXPHOS and regulate cell metabolism. The mitochondrial genome (mtDNA) and nuclear genome (nDNA) together control mitochondrial function, and when they are mutated, it may lead to mitochondrial dysfunction, such as excessive production of ROS and reduced OXPHOS capacity. Mitochondria, as a key place for cell metabolism to generate ATP, provide huge energy for the contraction and relaxation of human cardiac myocytes (HCM), and the accumulation of dysfunctional mitochondria will induce CVDs [6]. For example, in heart failure, the heart has a high demand for energy particularly and requires mitochondrial OXPHOS to support continuous ATP in cardiomyocytes [7].

In recent years, a growing number of studies have confirmed that mitochondrial dysfunction is a non negligible cause of CVDs. For instance, mtDNA mutation could disrupt mitochondrial homeostasis, produce oxidative stress, cause a rise in ROS levels, impair OXPHOS and damage energy metabolism, all of which are risk factors for CVDs. Thus, exploration of mtDNA mutation at the genetic level may be a highly advantageous therapeutic approach to identify and predict CVDs [8]. Dysfunctional mitochondria are removed by the autophagy-lysosomal system, however, hyper activation of mitophagy also leads to pathological conditions [9]. Mitochondrial oxidative capacity has been evaluated in relation to CVDs, and specific targeted antioxidant treatments that reduce ROS production and enhance ROS detoxification alleviate oxidative stress caused by mitochondria [10]. In brief, mitochondria can potentially be used as therapeutic targets for cardiovascular health interventions [11].

In this review, we mainly discuss and analyze the relationship between mitochondrial dysfunction (e.g., mtDNA mutations, impaired mitophagy, decreased OXPHOS and elevated ROS) and CVDs, and explore potential therapeutic strategies for CVDs by eliminating mtDNA mutations, enhancing mitophagy, improving OXPHOS capacity and reducing ROS. From the perspective of mitochondrial dysfunction, we aim to provide references for optimal treatment of CVDs.

## 2. Mitochondrial Dysfunctions and CVDs

The pathogenesis of CVDs is constantly being explored. Although the pathogenesis of CVDs is not fully understood; a study from the perspective of mitochondrial dysfunction and the analysis of the relationship between mitochondrial dysfunction and CVDs will help to better understand and solve the problems related to CVDs. Here, we summarize the relationships between primary CVDs and mitochondrial dysfunction (Table 1 and Figure 1), and further analyzed the four important features of mitochondria in CVDs in detail.
ijms-23-16053-t001_Table 1Table 1CVDs and corresponding mitochondrial dysfunction.CVDsMitochondrial DysfunctionReferencesHeart failureImpaired mitophagy↓OXPHOS↑ROS [10] review, 2019[12] experimental, 2010[13] experimental, 2011[14] review, 2022[15] experimental, 2018[16] experimental, 2020[17] experimental, 2011[18] experimental, 2015Myocardial hypertrophymtDNA mutationImpaired mitophagy↑ROS[10] review, 2019[17] experimental, 2011[19] experimental, 2022[20] experimental, 2019[21] experimental, 2021[22] experimental, 2021[23] experimental, 2015AtherosclerosismtDNA mutationImpaired mitophagy↓OXPHOS↑ROS[24] clinical, 2019[25] experimental, 2017[26] experimental, 2018[27] experimental, 2017[28] experimental, 2019[29] experimental, 2021[30] experimental, 2014Aortic dissection↑ROS[31] experimental, 2016[32] experimental, 2020[33] experimental, 2022[34] experimental, 2021Aortic aneurysm↑ROS↓OXPHOS[32] experimental, 2020[35] experimental, 2020[36] experimental, 2018[37] experimental, 2022[38] experimental, 2006[39] experimental, 2015[40] experimental, 2013[41] clinical, 2020[42] clinical, 2021“↑” indicate a rise and “↓” indicate a decrease.

### 2.1. mtDNA Mutation in CVDs

Human mtDNA is a circular double stranded genome with a length of 16,569 bp and consists of 37 genes, which support aerobic respiration and the production of cellular energy through OXPHOS. Unlike nuclear DNA, mtDNA is not protected by histones and does not recombine, resulting in approximately 10–100-fold higher mutation rates [43]. mtDNA mutations include point mutations deletions, fragment deletion and large scale mtDNA rearrangements, which can directly impair OXPHOS [44,45]. A large number of mitochondrial diseases are rooted in mtDNA mutations [46]. Notably, many specific disease mutations in mtDNA have been observed to cause cardiomyopathy, suggesting that mtDNA encoded proteins play a vital role in mitochondrial function of the heart [47]. Heart failure, a complex clinical syndrome that represents the end result of CVDs with multiple etiologies, is a bioenergetic disease with severe mtDNA mutations and mitochondrial dysfunction [14,48]. The MRPL44-disorder causes problems with the translation of a partial protein participating in OXPHOS, and it is associated with the clinical manifestation of cardiomyopathy in infancy [49]. In a mouse model of myocardial infarction (MI), the knockdown of the mouse lncRNA-SNHG8 gene significantly suppressed cardiac tissue injury [50]. Myocardial hypertrophy is a common inherited CVDs, and cardiomyocytes are associated with abnormal mitochondrial structure and dysfunction of mitophagy clearance, which makes it impossible to maintain mtDNA and functional integrity [4]. Hypertrophic cardiomyopathy is the predominant pattern of cardiomyopathy in mtDNA diseases, observed in nearly 40% of patients [51]. The studies have evaluated an association between mtDNA mutations and maternally inherited essential hypertension (MIEH), and these mutations may be one of the pathological mechanisms causing MIEH [45,52]. In addition, mtDNA mutations are also associated with atherosclerosis [53]. For example, four mutation genes including m.A1555 G in the MT-RNR1 gene, m.C3256 T in the MT-TL1 gene, m.G12315A in the MT-TL2 gene and m.G15059A in the MT-CYB gene are associated with atherosclerosis [54].

### 2.2. Mitophagy Damage in CVDs

Autophagy plays a positive role in maintaining cellular homeostasis in most cardiovascular-derived cells (e.g., cardiomyocytes, VSMCs) [55], and mitophagy is a kind of selective autophagy [56]. Mitophagy is one of the mitochondrial quality control pathways, and it can control and remove damaged mitochondria in cells [57]. During mitophagy, the damaged mitochondria are sequestered by double membrane vesicles and eventually become hydrolyzed by lysosomes [58]. Therefore, if mitophagy is impaired, the accumulation of dysfunctional mitochondria increases, which may lead to abnormal cell function and CVDs. Reducing mitochondrial dysfunction and lipid accumulation by activating mitophagy can help prevent diabetic cardiomyopathy caused by high fat diet [59]. In contrast, in *BMAL1* deficient hESC-derived cardiomyocytes, impaired mitophagy is a key cause for the development of dilated cardiomyopathy [60]. In Hu’s study, constructing mice with overexpression of omentin1 demonstrated that omentin1 activated mitophagy to improve HF [61]. In cardiac ischemia-reperfusion injury associated with disturbed mitochondrial homeostasis, the casein kinase 2α amplifies cardiomyocyte death signals by inhibiting mitophagy [62]. Defective mitophagy in VSMCs affects the progression of atherosclerotic lesions and promotes an unstable phenotype [28]. In addition, mitophagy damage in endothelial cells leads to senescence and apoptosis during atherosclerotic thrombosis [57].

### 2.3. Mitochondrial OXPHOS Reduction in CVDs

Defects in the genes encoding the OXPHOS complex are responsible for triggering various diseases, especially those with high energy requirements [63]. Dysfunction of OXPHOS is considered as one of the main causes of CVDs [64]. In chronic HF patients, reduced succinyl-CoA levels in myocardial mitochondria cause decreased OXPHOS [65]. In a study of human thoracic aortic aneurysm tissue, when mitochondrial OXPHOS related gene expression is inhibited, although chromatin OXPHOS related genes are increased, the ATP production is still insufficient to maintain contractile activity in human aortic smooth muscle cells (HAoSMCs) [41]. In an another study evaluating the effect of NOTCH1 deletion on the contractile phenotype and mitochondrial dynamics of human HAoSMCs, NOTCH1 deficiency can cause mitochondrial dysfunction in HAoSMCs by reducing mitochondrial fusion, inducing loss of mitochondrial membrane potential, increasing ROS generation, insufficient ATP production, and accompany with an impaired contractile phenotype [42]. PGC-1β deficiency in heart suppresses OXPHOS gene expression, and it can inhibit the transition from pressure overload myocardial hypertrophy to heart failure by modulating PGC-1β activity [13]. miR-27b-3p is thought to be related to OXPHOS. When it is inhibited, OXPHOS is enhanced and inhibits cardiomyocyte hypertrophy [66]. It has been also reported that decreased mitochondrial respiration and OXPHOS damage in epicardial adipose tissue were associated with coronary atherosclerosis severity [24].

### 2.4. Mitochondrial-Derived ROS Increase in CVDs

Mitochondrial ROS production is closely related to the mitochondrial ETC and NADPH oxidase (NOX). In the process of mitochondrial electron transfer, complex I and complex III are the main sites of ROS generation [67,68]. NADPH acts as a substrate to generate ROS under the action of NOX. The NOX is rich in mitochondria, and under the combined action of ETC and NOX, ROS continuously accumulates [69]. As a toxic by-product, ROS can damage mitochondria and are involved in the pathomechanism of CVDs. In turn, damaged mitochondria induce a large amount of ROS to be released from adjacent mitochondria, which is known as ROS-induced ROS [70]. The increased mitochondrial ROS represents one of the pathogenic mechanisms for vascular diseases [71]. For instance, both Nox2 and Nox4 induce oxidative stress, and the resulting ROS is closely related to ischemia-reperfusion [72]. The degree of atherosclerosis is associated with mitochondrial DNA damage, which associated with increased mitochondrial ROS [73]. It is not difficult to accept that the high reactivity of ROS will break the antioxidant balance that results in increased oxidative modification of the arterial wall. Studies have also shown that increased mitochondrial ROS leads to an increase in apoptotic cells and promotes age-related atherosclerosis [26]. Increased ROS predisposes endothelial cells to mitochondrial dysfunction, vascular inflammation, and accumulation of oxidized low-density lipoprotein, contributing to atherosclerosis and possibly plaque rupture [74]. VSMCs, as the predominant medial effector cells in aortic dissections and aneurysms, are a key factor in AD development. Increased ROS activates multiple hypertrophic signaling kinases and transcription factors, leading to dissection by inducing VSMCs apoptosis through the release of matrix metalloproteinases [31,75].

## 3. Strategies for Targeting Mitochondria to Treat CVDs

Exploring the mechanism of mitochondrial dysfunction in vascular diseases is a challenge for developing strategies to target mitochondria in CVDs. Here, we have summarized the relationship between four important features of mitochondria (mtDNA mutations, impaired mitophagy, decreased OXPHOS, elevated ROS) and CVDs. In view of these four features, we further sorted out the development and hotspot treatment strategies in recent years, and analyze the advantages and limitations of different treatment strategies (Table 2), hoping to find effective and operable solutions for all kinds of CVDs.

### 3.1. mtDNA Mutation and Treatment in CVDs

#### 3.1.1. mtDNA Editing Therapy

Mitochondrial heterogeneity affects mtDNA stability through copy number alterations and point deletions [91]. Once mtDNA is cleaved and linearized, it is rapidly degraded [92]. By duplicating residual mtDNA, mtDNA can be repopulated to the original level. In general, mitochondrial gene editing may include four potential approaches: mitochondria targeted restriction endonuclease (RE) technology, zinc finger nuclease (ZFN) technology, transcription activator-like effector nuclease (TALEN) technology and CRISPR/Cas9 system. In fact, mtDNA editing is a promising therapeutic modality to treat heteroplasmic or mutant mtDNA diseases. Specific mtDNA was effectively eliminated in heart of mice by using a mitochondria targeted RE [78,93]. Mitochondrial-targeted ZFNs can selectively cleave and degrade pathogenic mtDNA bearing large scale deletions or point mutations [94]. An alternative TALEN has been developed to effectively reduce mutant mtDNA and elevate OXPHOS in cells [95]. TALEN was used to provide a cure for some mitochondrial diseases caused by mtDNA mutations by specifically cleaving and eliminating pathogenic mtDNA mutations [96]. Several studies have reported that CRISPR/Cas9 system mediated mtDNA editing [97,98], and the use of PNPase to target mitochondria and eliminate mtDNA pathogenic mutations is quite promising [99]. Notably, recent studies have found that Ddda-derived cytosine base editor (DdCBE) exhibits higher fidelity and can improve the accuracy of mtDNA [100,101,102]. Efficient and heritable modification of the mouse mitochondrial genome has been shown to be mediated by DdCBE, which is used to potentially generate mtDNA mutation models in humans. This approach could theoretically reduce disease-causing mutational burdens below a threshold, and is a potential strategy to target mtDNA for the treatment of CVDs due to mitochondrial dysfunction and mtDNA mutations.

#### 3.1.2. Mitochondrial Replacement Therapy

mtDNA replacement therapy (MRT) is to use enucleated donor embryos as healthy mtDNA to replace undesired defective/mutated mtDNA to prevent mitochondria from being maternally inherited. MRT is a form of in vitro fertilization (IVF) that includes spindle transfer (ST), prokaryotic transfer (PNT) and polar body transfer (PBT) [103]. In fact, embryos from human nuclear transfer can contain low levels of mutated mtDNA, which may be suitable for treating degenerative diseases caused by mtDNA mutations [104]. This opens up the possibility of MRT for CVDs, a chronic noncommunicable degenerative disease. Hyslop et al. developed a PNT protocol that promotes efficient development at the blastocyst stage, keeping mtDNA residues as low as possible [105]. At present, there have been successful cases of applying MRT strategies [106,107], and offspring will not suffer from mtDNA mutation-related diseases. Therefore, hypertrophic cardiomyopathy, dilated cardiomyopathy, genetically related coronary heart disease and other CVDs can be considered using MRT, as an auxiliary means of human reproduction, to solve the problem from the embryo. However, the scientific knowledge related to MRT is still being explored, and the risks and ethical issues of this technology remain to be resolved [108].

### 3.2. Mitophagy Therapy in CVDs

Mitophagy clears dysfunctional mitochondria under normal physiological conditions, and in response to pathological stress [15]. Currently, there are three mechanisms of mitophagy, including mitochondrial outer membrane receptor-mediated, Pink1/Parkin pathway, and lipid receptor-mediated mechanisms (Figure 2) [109]. Notably, the mechanism mediated by the Pink1/Parkin pathway is the most extensively studied. Under normal circumstances, the content of Pink1 is extremely low. When oxidative stress occurs and mitochondria damage is induced, Pink1 is activated and recruits Parkin to the mitochondrial outer membrane for phosphorylation. The phosphorylated Parkin ubiquitinates the substrate protein on the mitochondrial membrane [110]. These ubiquitinated proteins subsequently recruit specific autophagy-related receptors to interact with LC3-II to form autophagosomes [111].

Mitophagy maintains cardiovascular homeostasis and performs significant functions in mitochondrial quality control. It has been shown that phosphorylation of Ser495 in Pink1 by AMPKα2 is necessary for effective mitotic inhibition of the progression of heart failure [15]. Ophiopogonin D’ (OPD’) is toxic to mitochondria, and OPD’-induced mitosis and mitochondrial damage in cardiomyocytes are partly mediated by the dysregulation of the Pink1/Parkin pathway, preventing excessive mitochondrial autophagy [112]. In a study on the improvement of cardiac function by berberine, it was found that the coordinated action of berberine and the Pink1/Parkin pathway enhances mitochondrial phagocytosis and protects patients with heart failure [16]. Ulk1/Rab9-dependent alternative mitophagy is activated during chronic high-fat diet depletion as an important mitochondrial quality control mechanism to protect the heart from the obesity effects of cardiomyopathy [81]. In conclusion, the control of mitophagy has an important role in the clearance of abnormal mitochondria and the protection of cardiomyocytes (Figure 2).

In addition, when abnormal mitochondria undergo fission, they can trigger cardiovascular dysfunction [113,114]. Cytoplasmic GTPase dynamics related protein 1 (Drp1) regulates mitochondrial fission by interacting with proteins located at fission sites such as mitochondrial fission 1 (Fis1), mitochondrial fission factor (Mff), and mitochondrial dynamics proteins of 49 and 51 kDa (MiD49 and MiD51) [115]. A study identified mitochondrial fission inhibitor (mdivi-1) as a cell-permeable quinazolinone derivative inhibitor of Drp1 [116]. In cardiomyocytes treated with mdivi-1, proteolytic cleavage of the OPA1 isoform and decreased expression of Mfn2, altered complex I and complex II protein expression of OXPHOS, and increased superoxide production were observed, which resulted in mitochondrial respiration defects and macro-autophagy inhibition [117]. Taken together, it is speculated that targeting mitochondrial fission or Drp1 may be useful for CVDs therapy.

### 3.3. Mitochondrial OXPHOS Reduction and Treatment in CVDs

#### 3.3.1. Small Molecule Compounds Enhance Mitochondrial Function

SIRT3 is a mitochondrial protein deacetylase that regulates mitochondrial function and is considered as an emerging drug target for CVDs [118]. SIRT3 can make mitochondrial metabolic pathways and ROS detoxification activate, and increase ATP production [119]. Resveratrol improves mitochondrial OXPHOS in diabetic hearts and prevents the decline of SIRT3 activity in the heart by increasing ETC activity and mitochondrial function [120]. The polyphenolic compound polydatin can initiate SIRT3-regulated mitophagy to prevent MI [84]. Notably, proteolytic targeting chimera technology, as a new strategy of targeted inhibitors, makes it possible to potently target small molecule compounds to enhance mitochondrial function, which may be more beneficial to the treatment of CVDs caused by mitochondrial dysfunction.

When mtDNA is damaged at high levels, increased Poly(ADP-ribose) polymerase (PARP) activity leads to a decrease in NAD^+^ levels, resulting in impaired NAD^+^-dependent SIRT3 activation and ultimately cardiac mitochondrial dysfunction [121]. Therefore, targeted improvement of mitochondrial function through nutritional supplementation NAD^+^ or ketoesters may be useful in patients with heart failure [87]. NAD^+^ supplementation with nicotinamide riboside (NR) promotes mitophagy in a Pink1-dependent manner [122]. NR can reduce ROS production and maintain normal mitochondrial function in the presence of inflammatory triggers [123]. The effect of nicotinamide mononucleotide (NMN) on the generation of ROS was investigated and it was finally found that NMN can reduce mitochondrial oxidative stress in brain microvascular endothelial cells and improve primary cerebro-microvascular endothelial cell membrane potential and mitochondrial respiration in a sirtuin-dependent manner [124]. Similarly, NMN improves the aorta by reducing oxidative stress [125]. Taken together, it can be seen that NR, NAD^+^, and NMN have certain therapeutic potential in the treatment of CVDs caused by mitochondrial dysfunction. Among them, NR and NMN still require further preclinical and clinical studies to ensure the safety of the drug [83].

#### 3.3.2. Nanomaterials Targeted Mitochondria to Improve Mitochondrial Function

Many drugs cannot precisely bind to damaged mitochondria, and they are even toxic to other tissues in the body. To solve these problems, precise targeted therapy has attracted much attention. Modification of nanoparticles with different components facilitates mitochondrial directed drug penetration [126]. A team has constructed a non invasive aerosol inhalation delivery system based on antioxidant nano drugs, which can target damaged mitochondria, clear ROS, and improve the targeting ability of nano-drugs to myocardium [85]. Artificial hybrid nanozymes created by protein reconstruction technology and nanotechnology can target mitochondria and scavenge ROS, thereby reducing mitochondrial oxidative damage [127]. Improved formulation of negatively charged peptide nanoparticles enables efficient localization of the drug to mitochondria [128]. Therefore, the novel nano drug delivery system in the human body to effectively treat human CVDs by targeting mitochondria will bring another bright future.

### 3.4. Reduction or Elimination of Mitochondrial-Derived ROS in CVDs

ROS acts as highly active molecules in vivo, and antioxidants can effectively reduce or eliminate ROS. The in vitro hypoxia/reoxygenation model of H9c2 cells could simulate myocardial ischemia-reperfusion injury, and it found that the experimental group supplemented with vitamin D could inhibit the production of ROS in cardiomyocytes [129]. Melatonin is an indole heterocyclic compound produced by pineal cells in the pineal gland. It can effectively lower ROS production, thereby reducing oxidative stress and VSMC loss, preventing the deterioration of thoracic aortic aneurysm and dissection [32]. Fullerenol nanoparticles are introduced into an alginate hydrogel to form a fullerenol/alginate hydrogel with antioxidant activity. This injectable cell delivery vector can treat myocardial infarction by effectively reducing ROS levels [130]. Cardioprotection of tetrahedral DNA nanostructures can significantly decrease oxidative stress and play a positive role in protecting against myocardial ischemia-reperfusion injury [131]. However, the clinical effects of ROS scavengers in CVDs are not always significant, probably because antioxidants can indiscriminately remove some physiological ROS. Therefore, finding drugs to target damaged mitochondria will improve the clearance of pathological ROS.

Fortunately, the antioxidant CoQ10 has been used in the clinical treatment of CVDs and has good curative effect. Ubiquinone, the oxidized form of CoQ10, transports electrons in the mitochondrial ETC and plays a crucial role in mitochondrial energy production. CoQ10 can transport H^+^ to thermally dissipate chemosmotic gradients via uncoupling proteins (UCP-1, 2 and 3). After uncoupling, the reduction level of electron carriers is reduced, thereby reducing the production of ROS [132]. Moreover, the reduced form of CoQ10 is also an active agent involved in antioxidant function, which can scavenge ROS production due to mitochondrial dysfunction [133]. Meanwhile, CoQ10 helps recycle other antioxidants such as radical forms of vitamin C and vitamin E [134]. CoQ10 has been shown to increase ATP production in cardiomyocytes, enhance oxidative effects, and improve endothelial function and lipid profile [135]. Comparing CoQ10 with placebo, the therapeutic effect of CoQ10 was more significant in the long term [136]. In addition, substantial clinical evidence suggests that CoQ10 supplementation (≥200 mg/day) contributes to cardiac health in patients affected by coronary heart disease and heart failure [90]. The safety profile of CoQ10 can be used as adjunctive therapy in congestive heart failure and may be helpful in patients who cannot tolerate mainstream drugs [137]. CoQ10 supplementation is safe and well tolerated with few drug interactions and side effects [138]. Similarly, the MitoQ was clinically demonstrated for its antioxidant effects on mitochondrial-derived ROS. The MitoQ can increase the resistance of aging mice to mitochondrial-derived ROS and protect against the imbalance of mitochondrial homeostasis due to aging. It is a novel strategy to treat and prevent age-related CVDs [139]. Of course, the task of applying more safe and effective new antioxidants to the clinical treatment of CVDs is a long way to go and needs to be continuously explored.

## 4. Conclusions and Perspective

As an important component of the cell, mitochondria contain genetic material, produce energy, and participate in a wide variety of metabolic activities in the cell. It can be seen that if the mitochondrial dysfunction occurs, the normal replication of mtDNA, energy production, and other functions will be affected, which may cause diseases. Here, we mainly analyzed the relationship between the heart and arterial-related CVDs, and mitochondrial dysfunction. Mitochondrial dysfunction in cardiomyocytes, vascular smooth muscle cells, and endothelial cells causes a wide variety of CVDs; and has attracted more and more scientists. With the deepening of CVDs pathogenesis related studies, we summarized the mitochondrial dysfunction causing CVDs into four important characteristics, including mtDNA mutations, impaired mitophagy, decreased OXPHOS, and mitochondrial-derived ROS increase. In multiple animal and human models, many relevant intervention experiments have been designed according to mitochondrial dysfunction, constantly exploring more effective CVDs related therapeutic strategies. According to the four important characteristics of mitochondrial dysfunction, the related treatment strategies of CVDs were sorted out. Exploring the significance, advantages, and current limitations of different mitochondrial targeted therapy strategies can provide more ideas and options for the treatment of different CVDs. Although each of these strategies for ameliorating mitochondrial dysfunction has its own characteristics, combination therapy may be more effective. It is well known that CVDs are quite complex, and their pathological mechanisms are even more complex and diverse. On the basis of continuously deepening the pathological mechanisms, mitochondrial targets can be found more accurately. In the case of harmless to human body, mitochondrial targeted therapy for CVDs may improve the efficiency and safety of treatment, and contribute to the development of human health.

## Figures and Tables

**Figure 1 ijms-23-16053-f001:**
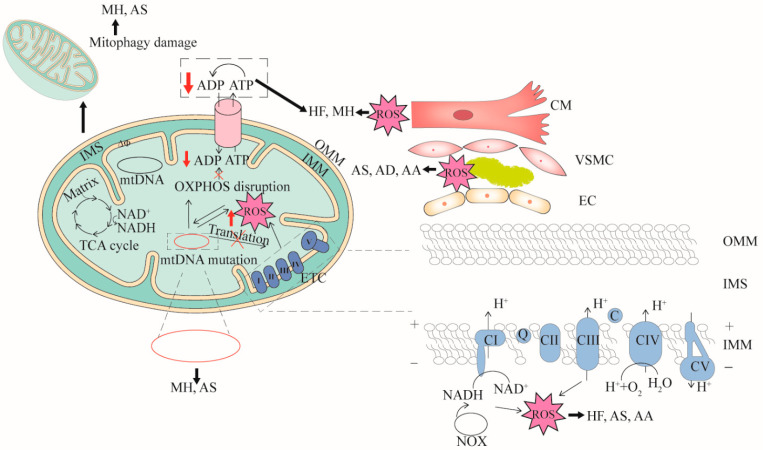
The relationships between mitochondrial dysfunction and CVDs. Four features (mtDNA mutation, mitophagy damage, decreased OXPHOS and increased ROS) associated with mitochondrial dysfunction are demonstrated. mtDNA mutation can cause dysfunction of mitochondrial respiratory chain complex or cytochrome transcription related to OXPHOS. When OXPHOS is impaired, ATP synthesis is reduced and excess ROS is generated. mtDNA mutation directly affects mitochondrial function or ROS production. In turn, high levels of ROS damage mitochondria. Cardiomyocyte cells (CM), vascular smooth muscle cells (VSMC), and endothelial cells (EC) that are impaired by mitochondrial dysfunction can cause CVDs. mtDNA mutation can cause myocardial hypertrophy (MH) and atherosclerosis (AS), mitophagy damage can cause MH and AS, decreased OXPHOS can cause heart failure (HF), aortic aneurysm (AA) and AS, and increased ROS can cause AS, aortic dissection (AD) and AA. OMM, outer mitochondrial membrane; IMS, inter membrane space; IMM, inner mitochondrial membrane.

**Figure 2 ijms-23-16053-f002:**
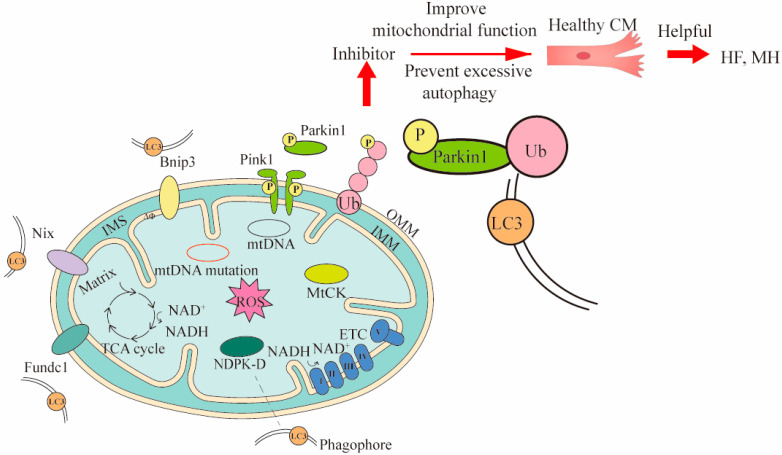
Three mechanisms of mitophagy and the ways they intervene in treating CVDs. Three mechanisms of mitophagy include mitochondrial outer membrane receptor-mediated (such as Bnip3, Nix and Fundc1), Pink1/Parkin pathway, and lipid receptor-mediated mechanisms (such as MtCK, NDPK-D). The LC3 is located in the phagophore and binds to the corresponding receptor. The LC3 can bind to substances of different mitophagy mechanisms. To demonstrate the mitophagy occurring in CVDs, cardiomyocytes (CM) and Pink1/Parkin pathway were used to intervene in heart failure (HF) and myocardial hypertrophy (MH) by inhibitors. In Pink1/Parkin-dependent mitophagy, the Pink 1 accumulated on the damaged mitochondria is activated and recruits Parkin for phosphorylation. The phosphorylated Parkin binds to the ubiquitin attached to outer OMM, and finally binds to LC3 for mitophagy. The Bnip3 and Nix can directly bind LC3 and promote mitophagy. MtCK and NDPK-D as specific transporters can also directly bind LC3 for mitophagy to eliminate damaged mitochondria. The inhibitor acts on Pink1/Parkin pathway, and then prevents excessive mitophagy and improves mitochondrial function, thereby maintaining the healthy levels of cells associated with CVDs. OMM, outer mitochondrial membrane; IMS, inter membrane space; IMM, inner mitochondrial membrane.

**Table 2 ijms-23-16053-t002:** Advantages and limitations of treatment strategies for targeting mitochondria to treat CVDs.

Features	Strategies	Advantages	Limitations	Examples as Applied	References
Mutant mtDNA	mtDNA editing	High specificity; Easy operation	High cost; Limited availability	mito-RE, heart related CVDs	[46,76,77,78]
	Mitochondrial replacement therapy	Reduce the risk of vertical transmission	Ethical and legal issues		[79]
Impaired mitophagy	Enhanced mitophagy	Play protective roles	Unclear conditions for mitophagy; A potential cytotoxic	Berberine and the Pink1/Parkin pathway, HF; Active Ulk1/Rab9-dependent, Cardiomyopathy	[16,55,80,81]
Decreased OXPHOS	Small molecule compounds to improve OXPHOS	Dietary supplements; Wide range of sources	Lack of clinical trials; Being degraded in advance	Control of the SIRT3 activity, MI	[82,83,84]
	Nanomaterials to enhance mitochondrial function	Precise targeting; Noninvasive; High load drug	Lack of clinical trials; Collaborative targeting against multiple subcellular organelles is limited	NAD^+^, HF	[85,86,87]
Increased ROS	Antioxidant	Amounts of clinical trials (CoQ10)	Interaction with statin (CoQ10); Long-term exposure maybe harmful	CoQ10, HF; Melatonin, AA/AD	[32,88,89,90]

HF, heart failure; MI, myocardial infarction; AA, aortic aneurysm; AD, aortic dissection; SIRT3, Sirtuin-3.

## Data Availability

Not applicable.

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
