# Peer review of "Mitochondrial Dysfunction and Therapeutic Perspectives in Cardiovascular Diseases"

_ijms, 2022, doi:10.3390/ijms232416053_

Round 1

Reviewer 1 Report

The review work entitled "mitochondrial dysfunction and therapeutic perspectives in cardiovascular diseases" is comprehensive collection of work done to explore the possible role of mitochondria in pathogenesis of CVD's and treatment options targeting mitochondria. 

The review is interesting and informative and provide in detail information about role of mitochondria in CVD's. 

I have followings comments and suggestions which would improve the quality of work presented. 

Abstract:

line 9: I suggest to modify " rate of" to "rate due to" 

Line 14: change discuss and analyze to discussed and analyzed 

Line 16: modify explore to explored

in key words in my opinion we can add full word instead of abbreviation

Introduction 

Line 35: 20% of all age related death. I assume this % of death is related to death due to all CVD which are mainly coronary artery disease and cardiac arrest but not CVD due to mitochondria. If you have % of death mentioning CVD's related death due to mitochondrial dysfunction please mention here. 

Line 43: please rewrite this sentence to be more clear 

Table 1: the table is very superficial, since this is review paper i suggest to add columns with details about type of study (clinical, experimental, review), year of publication and any other important detail.

Line 87: replace the word In contrast with "Unlike"

Line 88: remove "in mtDNA than in nuclear DNA"

Table 2: I suggest to add column related to CVD which could be treated using mentioned strategies.

Other sections are clear and understandable. Images are good 

Good luck 

Author Response

Comments for Transmission to Authors

The review work entitled "mitochondrial dysfunction and therapeutic perspectives in cardiovascular diseases" is comprehensive collection of work done to explore the possible role of mitochondria in pathogenesis of CVD's and treatment options targeting mitochondria.

The review is interesting and informative and provide in detail information about role of mitochondria in CVD's.

I have followings comments and suggestions which would improve the quality of work presented.

RESPONSE: We sincerely thank the reviewer for the encouraging and insightful comments.

Abstract:

line 9: I suggest to modify " rate of" to "rate due to"

RESPONSE: Thank you and we have modified.

Line 14: change discuss and analyze to discussed and analyzed

RESPONSE: Thank you and we have changed.

Line 16: modify explore to explored

RESPONSE: Thanks and we have modified “explore” to “explored”.

in key words in my opinion we can add full word instead of abbreviation

RESPONSE: Thanks and we have added full word instead of abbreviation.

Introduction

Line 35: 20% of all age related death. I assume this % of death is related to death due to all CVD which are mainly coronary artery disease and cardiac arrest but not CVD due to mitochondria. If you have % of death mentioning CVD's related death due to mitochondrial dysfunction, please mention here.

RESPONSE: Thanks to reviewer's suggestion and we agree with the opinion. However, the % of death mentioning CVD's related death due to mitochondrial dysfunction cannot be found.

Line 43: please rewrite this sentence to be more clear

RESPONSE: Thank you and we have rewritten.

Table 1: the table is very superficial, since this is review paper i suggest to add columns with details about type of study (clinical, experimental, review), year of publication and any other important detail.

RESPONSE: We thank the reviewer for this suggestion. We have added columns with details about type of study (clinical, experimental, review) and year of publication.

Line 87: replace the word In contrast with "Unlike"

RESPONSE: Thanks and we have revised.

Line 88: remove "in mtDNA than in nuclear DNA"

RESPONSE: Thanks and we have revised.

Table 2: I suggest to add column related to CVD which could be treated using mentioned strategies.

RESPONSE: Thank you for your suggestion and we have added a column in Table 2.

Other sections are clear and understandable. Images are good.

RESPONSE: Thank you for your encouragement.

Reviewer 2 Report

This review focuses on the role of mitochondrial dysfunctions in cardiovascular diseases (CVDs) and therapeutic strategies targeted on mitochondria. In the first part, only brief overview of mitochondrial dysfunctions in CVDs is presented. Therapeutic strategies are discussed in more detail and authors present interesting information on significance, advantages and limitations of treatment strategies targeted on different aspects of mitochondrial dysfunctions. The manuscript is generally well written and presents important information, mainly on therapies targeting mitochondria, however, there are several points that should be addressed and need attention.

Line 119: “with mitochondrial homeostasis“ should be: with disturbed mitochondrial homeostasis.

Line 14: “    and even cause CVDs” better should be:    and are involved in the pathomechanism of CVDs.

Line 251: “GTpase” should be GTPase.

Line 263 and Table 2: “molecule compounds” is nonsense, possibly should be: small molecule compounds.

Lines 267-269: “Resveratrol ……..”, the reference is missing.

Line 269: “polygonin” should be polydatin.

Lines 278-280: Please rewrite the sentence “Mitophagy can also      “. Maybe it could be: NAD+ supplementation with NR promotes mitophagy in a Pink1-dependent manner.

Line 304: Please, change the text: “ROS is one of the common free radical in the body,”. ROS is a general term.

Line 307: Paper 101 does not deal with vitamin D associated reduction of ROS levels. Moreover, antioxidant role of vitamin D is questionable.

Lines 347-348: “Whether some CVDs are associated with mitochondrial dysfunction remains unclear …”. This seems to be incorrect, moreover, in several passages the authors claim the opposite, see lines 37-38, 47-48, 126-127.

Section 3.4.: Studies on other antioxidants, such as mitoQ, should be included.

Author Response

This review focuses on the role of mitochondrial dysfunctions in cardiovascular diseases (CVDs) and therapeutic strategies targeted on mitochondria. In the first part, only brief overview of mitochondrial dysfunctions in CVDs is presented. Therapeutic strategies are discussed in more detail and authors present interesting information on significance, advantages and limitations of treatment strategies targeted on different aspects of mitochondrial dysfunctions. The manuscript is generally well written and presents important information, mainly on therapies targeting mitochondria, however, there are several points that should be addressed and need attention.

RESPONSE: We sincerely thank the reviewer for the encouraging and insightful comments.

Line 119: “with mitochondrial homeostasis” should be: with disturbed mitochondrial homeostasis.

RESPONSE: Our apology and corrected.

Line 14: “and even cause CVDs” better should be:  and are involved in the pathomechanism of CVDs.

RESPONSE: Thanks and revised.

Line 251: “GTpase” should be GTPase.

RESPONSE: Our apology and corrected.

Line 263 and Table 2: “molecule compounds” is nonsense, possibly should be: small molecule compounds.

RESPONSE: Thanks and revised.

Lines 267-269: “Resveratrol ……..”, the reference is missing.

RESPONSE: Our apology and revised.

Line 269: “polygonin” should be polydatin.

RESPONSE: Our apology and corrected.

Lines 278-280: Please rewrite the sentence “Mitophagy can also”. Maybe it could be: NAD+ supplementation with NR promotes mitophagy in a Pink1-dependent manner.

RESPONSE: Thank you for your suggestion and we have rewritten the sentence.

Line 304: Please, change the text: “ROS is one of the common free radical in the body,”. ROS is a general term.

RESPONSE: Thanks and we have changed.

Line 307: Paper 101 does not deal with vitamin D associated reduction of ROS levels. Moreover, antioxidant role of vitamin D is questionable.

RESPONSE: Thank you and we have revised the Reference and description about vitamin D associated reduction of ROS levels. In addition, the description of the antioxidant role of vitamin D is deleted.

Lines 347-348: “Whether some CVDs are associated with mitochondrial dysfunction remains unclear …”. This seems to be incorrect, moreover, in several passages the authors claim the opposite, see lines 37-38, 47-48, 126-127.

RESPONSE: Thank you for pointing out this issue. We have removed this sentence from Lines 347-348 to prevent inconsistencies.

Section 3.4.: Studies on other antioxidants, such as mitoQ, should be included.

RESPONSE: Thanks and we have added mitoQ related content in Section 3.4.

Reviewer 3 Report

Dear Author

This manuscript regarding "Mitochondrial Dysfunction and Therapeutic Perspectives in 2 Cardiovascular Diseases" is relatively well-written. 

Here are the major comments. 

1. In a search of PubMed, there are about 4000 papers about "Mitochondrial Dysfunction and Cardiovascular Diseases" within 5 years. The contents of each sub-session 2 are too short. Thus, please, enrich the contents of each session by adding additional references except for session 3. 

2. The contents of the conclusion and Perspective are also too short. Please, complement this part more.

3. Table 2 is too broad. Thus, fulfill the detailed experimental methods from the corresponding references (60~). 

4. This manuscript has too many reviews. Please, replace some review papers with articles. 

Here are minor comments. 

1. Please, change "Increased or decreased" with the arrow in Table1. 

2. Please, the figure images are not so good, and clarify the image of Figures 1, and 2

Author Response

This manuscript regarding "Mitochondrial Dysfunction and Therapeutic Perspectives in 2 Cardiovascular Diseases" is relatively well-written. 

RESPONSE: Thank you for your encouraging comments.

Here are the major comments. 

  1. In a search of PubMed, there are about 4000 papers about "Mitochondrial Dysfunction and Cardiovascular Diseases" within 5 years. The contents of each sub-session 2 are too short. Thus, please, enrich the contents of each session by adding additional references except for session 3. 

RESPONSE: Thank you for this suggestion and we have added additional references to enrich the contents of each sub-session 2.

  1. The contents of the conclusion and Perspective are also too short. Please, complement this part more.

RESPONSE: Thanks for pointing out this issue. We have enriched the contents of the conclusion and perspective.

  1. Table 2 is too broad. Thus, fulfill the detailed experimental methods from the corresponding references (60~). 

RESPONSE: Thanks and we have added a column in Table 2 to fulfill the detailed experimental methods and application in CVDs.

  1. This manuscript has too many reviews. Please, replace some review papers with articles. 

RESPONSE: Thank you and we have replaced some reviews with experimental articles and also supplemented some experimental articles.

Here are minor comments. 

  1. Please, change "Increased or decreased" with the arrow in Table1. 

RESPONSE: Thanks and we have changed.

  1. Please, the figure images are not so good, and clarify the image of Figures 1, and 2

RESPONSE: Thank you for pointing out this issue and we have revised figure legends of Figures 1 and 2.

Round 2

Reviewer 1 Report

The manuscript is revised as per the suggestions and comments raised.

Please consider the following minor corrections.

Line 37: kindly remove "of all age-related deaths" as it doesn't make any significant impact and moreover death word is already mentioned in previous sentence. 

Table 2: correct the spelling mistake in "heart raleted"

Table 2: provide the full name below the table for abbreviations mentioned in table such as HF, MI, AA/AD, SIRT3. This would make the readers to understand the table easily. 

Author Response

The manuscript is revised as per the suggestions and comments raised.

RESPONSE: Thank you for your encouraging comments.

Please consider the following minor corrections.

Line 37: kindly remove "of all age-related deaths" as it doesn't make any significant impact and moreover death word is already mentioned in previous sentence.

RESPONSE: Thanks and we have removed.

Table 2: correct the spelling mistake in "heart raleted"

RESPONSE: Our apology and corrected.

Table 2: provide the full name below the table for abbreviations mentioned in table such as HF, MI, AA/AD, SIRT3. This would make the readers to understand the table easily.

RESPONSE: Thank you and we have added the full name below the table for abbreviations.

Reviewer 3 Report

Dear Authors 

This revised manuscript is much better than before. 

Here are minor comments. 

1. Please, put the arrow (up or down) in front of the sentence in the table. 

2. The images of Figures 1 and 2  are still the same. Please, clarify the text (for example, enlarge the font size), and enlarge the size of the figure. 

3. Each symptom in table 1 is matched with only one experimental reference. Please, include at least 2-3 references (Only review is not acceptable.)

Author Response

This revised manuscript is much better than before.

RESPONSE: We sincerely thank the reviewer for the insightful comments.

Here are minor comments.

  1. Please, put the arrow (up or down) in front of the sentence in the table.

RESPONSE: Thanks and we have revised.

  1. The images of Figures 1 and 2 are still the same. Please, clarify the text (for example, enlarge the font size), and enlarge the size of the figure.

RESPONSE: Thank you and we have enlarged the size of the font and the figure.

  1. Each symptom in table 1 is matched with only one experimental reference. Please, include at least 2-3 references (Only review is not acceptable.)

RESPONSE: Thanks and we have added more relevant references to each symptom in Table 1.